# Assessing repetitive negative thinking in daily life: Development of an ecological momentary assessment paradigm

**Tabea Rosenkranz**[1]*, **Keisuke Takano**[1], **Edward R. Watkins**[2], **Thomas Ehring**[1]

**1** Department of Psychology, LMU Munich, Munich, Germany, **2** Mood Disorders Centre, School of Psychology, University of Exeter, Exeter, United Kingdom

* tabea.rosenkranz@psy.lmu.de

**Data Availability Statement:** All relevant data are available from the Open Science Framework database at DOI: 10.17605/OSF.IO/QRFG6.

## Abstract

Repetitive negative thinking (RNT) is a transdiagnostic process and a promising target for prevention and treatment of mental disorders. RNT is typically assessed via self-report questionnaires with most studies focusing on one type of RNT (i.e., worry or rumination) and one specific disorder (i.e., anxiety or depression). However, responses to such questionnaires may be biased by memory and metacognitive beliefs. Recently, Ecological Momentary Assessment (EMA) has been employed to minimize these biases. This study aims to develop an EMA paradigm to measure RNT as a transdiagnostic process in natural settings. Based on empirical and theoretical considerations, an item pool was created encompassing RNT content and processes. We then (1) tested model fit of a content-related and a process-related model for assessing RNT as an individual difference variable, (2) investigated the reliability and construct validity of the proposed scale(s), and (3) determined the optimal sampling design. One hundred fifty healthy participants aged 18 to 40 years filled out baseline questionnaires on rumination, worry, RNT, symptoms of depression, anxiety, and stress. Participants received 8 semi-random daily prompts assessing RNT over 14 days. After the EMA phase, participants answered questionnaires on depression, anxiety, and stress again. Multilevel confirmatory factor analysis revealed excellent model fit for the process-related model but unsatisfactory fit for the content-related model. Different hybrid models were additionally explored, yielding one model with satisfactory fit. Both the process-related and the hybrid scale showed good reliability and good convergent validity and were significantly associated with symptoms of depression, anxiety, and stress after the EMA phase when controlling for baseline scores. Further analyses found that a sampling design of 5 daily assessments across 10 days yielded the best tradeoff between participant burden and information retained by EMA. In sum, this paper presents a promising paradigm for assessing RNT in daily life.

## Introduction

Repetitive negative thinking (RNT) is a transdiagnostic process that has been shown to play an important role in the development and maintenance of emotional disorders [1–3]. RNT can

**Funding:** This project has received funding from the European Union's Horizon 2020 research and innovation program under grant agreement No 754657. ERW and TE have been awarded the funding (https://ec.europa.eu/programmes/horizon2020/en). The funders had no role in study design, data collection and analysis, decision to publish, or preparation of the manuscript.

**Competing interests:** The authors have declared that no competing interests exist.

be defined as a recurrent thinking process that is focused on negative content and perceived as difficult to control. Thus, RNT encompasses different phenomena that have traditionally been studied in isolation, such as worry, depressive rumination, and post-event processing in social anxiety [1]. Recent treatment approaches targeting RNT have yielded promising results, indicating that RNT can be regarded as a proximal risk and maintaining factor that is a suitable target for both prevention and treatment [e.g., 4–6, see also 7]. Due to the relevance of RNT as a transdiagnostic factor, it is of crucial importance that reliable and valid measures of RNT are available for both research and clinical purposes.

So far, RNT is commonly measured as a *trait* using self-report questionnaires, i.e. asking respondents about the typical content and/or style of their thinking [8–10]. However, the validity of such questionnaires has been called into question. Researchers have argued that self-report questionnaires, especially when referring to a retrospective period (e.g., in the past month) or assessing typical behavior/experience from a global perspective (e.g., general tendencies or traits), reflect a constructed experience that is biased by time, (metacognitive) beliefs, and state factors, rather than reflecting the actual experience or behavior [cf. 11,12].

Discrepancies between retrospective or global reports and real-time assessments of actual experiences have repeatedly been shown by research. For instance, people tend to overestimate internal experiences, such as emotions and physical pain, when asked retrospectively [cf. 11, see 12, for a review]. In the context of RNT, one study found that trait measures of worry only accounted for 24% of the variance of daily-life worry [13]. Furthermore, a recent study showed that momentary rumination predicted higher cortisol levels both in depressed and healthy participants in daily life, whereas trait rumination and retrospectively assessed depressive symptoms failed to predict this effect [14].

Thus, real-time assessments represent a promising approach to capture momentary experiences and contextual information, while minimizing retrospective biases and retaining ecologically valid data [11,15]. Ecological momentary assessment [EMA; 16]–also referred to as the Experience Sampling Method [ESM; 17]–captures individual processes in (near) real-time across multiple time points in a natural environment.

In the recent decade, several studies have employed EMA to assess different forms of momentary RNT [14,18–21]. However, the majority of these studies have predominantly focused on one specific content of RNT only (e.g., either worry, i.e. thinking about potential negative future situations, or rumination, i.e., self-focus on problems and feelings) within the context of specific disorders or symptom dimensions (e.g., either anxiety or depression). To the best of our knowledge, no EMA studies to date have investigated RNT as a transdiagnostic construct by focusing not only on the content but also on the process of RNT across symptom dimensions. Furthermore, even though EMA has received much attention from researchers during the past years thanks to the rapid technological progress, most studies have employed items based on face validity. A systematic investigation into the psychometric properties of the EMA assessment is largely missing [22], which is also the case in the field of RNT. Finally, most EMA studies in this area have focused on assessing state rumination or worry in order to investigate its relationship with, e.g., state changes in mood, cognitions or physiology [e.g., 23,24]. In contrast, the current study aims to develop a brief EMA paradigm to reliably measure the tendency to engage in RNT in daily life as an individual difference variable. To this end, we followed three methodological steps: (1) determining the items of the EMA paradigm, (2) testing the reliability and construct validity of the EMA scale(s), and (3) determining the optimal sampling frequency and duration (i.e., sampling design) of the EMA paradigm. In sum, the present paper presents the development of a reliable and valid EMA paradigm to assess RNT as an individual difference variable in daily life.

## Materials and methods

### Participants

In total, 150 participants aged 18 to 40 years ($M$ = 22.46, $SD$ = 4.01, 66.8% female) filled out the baseline questionnaires. Participants were included in the study if they had German language skills comparable to a native speaker and were currently not in treatment for mental disorders, as this study aimed for a non-clinical sample. Participants received either course credit or monetary compensation. For monetary compensation, participants received 8 € per hour for appointments and had the chance to win one of four 50 € vouchers depending on their compliance during EMA. For student participants, course credit was also given based on compliance during EMA. Nine participants dropped out due to technical difficulties or personal reasons and one participant was excluded due to being in treatment for a mental disorder. The final sample included 140 individuals for further analysis.

### Self-report questionnaires

**Rumination, worry, and repetitive negative thinking.** Rumination was assessed using the Responses Styles Questionnaire (RSQ-10D) [25,26]. The RSQ-10 is based on the Ruminative Responses Scale (RRS) by Nolen-Hoeksema and Morrow [27], but focuses on items that are not confounded with current symptoms of depression. Items are rated from 1 ("never") to 4 ("almost always"). A brooding and a reflection subscale can be computed, each consisting of five items. As higher brooding scores have been shown to predict depression, this subscale is assumed to capture dysfunctional rumination [21,26,28]. The RSQ-10 shows high reliability with an internal consistency of Cronbach's α = .92 for all items, and .75 and .78 for reflection and brooding, respectively [28]. In the current study, internal consistency was α = .69 for the brooding scale.

The Penn State Worry Questionnaire (PSWQ) [29,30] was used to assess trait worry. The PSWQ consists of 16 items measuring pathological worry. Each statement is rated on a 5-point scale (1 = "not at all typical of me", 5 = "very typical of me"). After recoding five negatively worded items, a total score from 16 to 80 can be achieved with higher scores indicating higher worry levels. Internal consistency ranges from α = .87 to .93 in non-clinical samples and the PSWQ shows good convergent and discriminant validity [31–33]. In the current study, internal consistency was very good with α = .90.

To assess transdiagnostic process characteristics of RNT, the Perseverative Thinking Questionnaire (PTQ) [8] was used. The PTQ comprises three core criteria of RNT, namely repetitiveness, intrusiveness, and difficulty to disengage from negative thoughts. It also measures mental capacity occupied by RNT and subjective unproductiveness of RNT. In total, 15 items are rated from 0 ("never") to 4 ("almost always") with a total score ranging from 0 to 60. The PTQ possesses high internal consistency, ranging from α = .92 to .94, and good discriminant and convergent validity [8,34]. In the current study, internal consistency was α = .93.

**Stress, anxiety, and depression.** Stress was assessed using the Perceived Stress Scale (PSS-10) [35,36]. The scale measures the degree to which participants have perceived their life situations in the past month as stressful, more precisely as uncontrollable, overwhelming and unforeseeable. Higher scores reflect a higher level of perceived stress, as the items are rated from 0 ("never") to 4 ("very often"), resulting in a maximum score of 40. The 10-item version of the PSS shows a good reliability of α = .78 to .91 [37] and good concurrent validity [38]. In the current study, internal consistency was α = .84.

Generalized anxiety was assessed using the Generalized Anxiety Disorder questionnaire (GAD-7) by Spitzer, Kroenke, Williams, and Löwe [39]. The presence and severity of seven

GAD symptoms over the past two weeks are rated on a 4-point scale (0 = "not at all", 3 = "nearly every day"). This results in a sum score of 21, with a cutoff score for GAD of 10. Internal consistency ranges between α = .82 and .92 in German studies, reflecting high reliability [40,41]. The GAD-7 correlates adequately with depression and self-esteem scales, suggesting construct validity [42]. In the current study, internal consistency was somewhat lower with α = .78 for the GAD-7.

Depression was assessed using the Patient Health Questionnaire–Depression (PHQ-D), [43]. All nine DSM-IV criteria for Major Depressive Disorder are rated as 0 ("not at all") to 3 ("nearly every day") regarding their presence in the last two weeks. While the PHQ-D can be used to screen for depression, we used the sum score of the PHQ-D to assess severity of depressive symptoms on a continuum. The PHQ-D has excellent internal reliability, demonstrated by α = .86 to .89 [44]. In the current study, internal consistency was somewhat lower with α = .78 for the PHQ-D.

Additionally, stress, anxiety, and depression was assessed using the Depression-Anxiety-Stress-Scales (DASS), which are suitable for measuring these symptoms in non-clinical populations [45,46]. Forty-two items assess symptoms of depression, anxiety, and stress over the past week on three subscales with a 3-point rating (0 = "did not apply to me at all", 3 = "applied to me very much, or most of the time"). All scales possess good discriminant and convergent validity and high reliability with α = .91, .78 to .82, and .81 to .89 for depression, anxiety, and stress, respectively [45,47]. In the current study, internal consistency ranged from α = .70 to .79.

## Ecological Momentary Assessment (EMA)

Participants received 8 notifications per day, pseudorandomized over a 12-hour time-window starting from 10 AM on weekdays and 12 PM on weekends for 14 consecutive days. After receiving a notification, participants had 15 minutes to start answering 12–14 questions about current mood, emotional events since the last notification, and current content and processes of RNT (see Table 1). Participants were reminded 5 and 10 minutes after the first notification if they had not answered the questions up to this point.

**Table 1. EMA paradigm: Items assessing repetitive negative thinking.**

| RNT Aspect | Item | Scale |
|---|---|---|
| Content | | |
| 1. Feelings (FEEL) | At the moment I am thinking about my feelings | 1: "not at all", 7: "very much" |
| 2. Problems (PROB) | At the moment I am thinking about my problems | 1: "not at all", 7: "very much" |
| 3. Past(PAST) | At the moment I am thinking about upsetting memories | 1: "not at all", 7: "very much" |
| 4. Future (FUT) | At the moment I am thinking about negative future situations | 1: "not at all", 7: "very much" |
| Process | | |
| 5. Duration (DUR) | How long have you been thinking about these topics up to this moment? | 1: "not at all", 7: "more than 120 min." |
| 6. Distress (BUR) | How much do you feel weighed down by these thoughts at this moment? | 1: "not at all", 7: "very much" |
| 7. Repetitiveness (RPT) | The same thoughts keep going through my mind again and again. | 1: "not at all", 7: "very much" |
| 8. Intrusiveness (INTR) | Thoughts come to my mind without me wanting them to. | 1: "not at all", 7: "very much" |
| 9. Uncontrollability (CTRL) | I get stuck on certain issues and can't move on. | 1: "not at all", 7: "very much" |

**Momentary repetitive negative thinking.**   In order to determine the items for the EMA paradigm, a literature research was conducted to identify concepts and items used in earlier studies to assess RNT.

1. *Content.* Four items were used to assess the content of RNT. The first two items asked respondents to indicate how much they are currently thinking about their feelings and problems, respectively. The items were modeled after similar items used in earlier studies [21], but phrased as whole sentences to fit with the format of the EMA assessment in this study. A composite score of both items has been used in various studies, showing good reliability and validity [14,21,23,48,49]. Worry and rumination are the two most frequently studied forms of RNT, and have been shown to share the same processes while differing in their temporal orientation [2,50]. In order to represent both forms of RNT in the EMA scale, we further included two items (thinking about upsetting events/ negative future situations) modeled after similar items originally introduced by Selby and colleagues [20]. An average score of all four items was significantly associated with trait rumination. Within-person reliability for these items ranged from $R_{KF}$ = .89 to .95 [20]. These four items made up the content-related model.

2. *Process.* Whereas RNT has traditionally been measured by asking participants about the *content* of recurrent thinking (as also reflected in the EMA items described above), recent transdiagnostic concepts of RNT put a stronger emphasis on characteristics of thinking *process* during RNT that is suggested to be transdiagnostic, i.e., similar across diagnostic categories [1,51]. An alternative way to measure RNT, therefore, focuses on assessing these process characteristics. For example, the Perseverative Thinking Questionnaire [8] was developed as a trait questionnaire for this purpose. In this study, three items were adapted from the PTQ to assess repetitiveness, intrusiveness, and difficulty to disengage from negative thoughts as the core characteristics of RNT. The latter, also labeled as uncontrollability, has been used in other EMA studies showing a significant association with trait rumination and prolonged sleep onset latency [52,53]. All three items showed significant factor loadings between .81 and .85 onto one factor in the validation study. Amongst others, this factor was significantly related to the brooding subscale of the RSQ, pathological worry, as well as symptoms of depression and anxiety [8,54,55].

Furthermore, it is assumed that both subjective burden and duration of RNT capture maladaptive processes as they indicate distress and failure to stop negative thoughts [13,18,56,57]. Therefore, one item assessing distress (subjective burden) caused by RNT and one item assessing duration of RNT were adapted from the work of Thielsch and colleagues [18]. The latter item was designed to assess duration of RNT between the previous and current notification, as participants received a notification about every two hours.

In addition to RNT items, participants also answered EMA questions regarding current mood and emotional events experienced, which are not relevant for the analyses of this paper.

## Procedure

Participants were recruited via posters, online announcements in social media and mailing lists. At the first appointment, the investigator explained the purpose and procedure of the study. After providing informed consent, participants were introduced to the EMA app ("Tellmi", developed at LMU Munich for research use), which they had installed on their smartphone (iOS or Android) prior to the appointment. If they did not own a smartphone, or had technical difficulties with the app on their smartphone, participants received a smartphone for the duration of the study. Participants were given the opportunity to look at all EMA

questions and ask questions regarding app use and procedure. Finally, participants filled out demographic information and the PHQ-D, GAD-7, DASS, PSS, PSWQ, RSQ-10, and PTQ. The EMA period started on the day after the first appointment and lasted 14 days. After this period, participants had a second appointment to again fill out the PHQ-D, GAD-7, DASS and PSS regarding the past weeks. Additionally, a semi-structured interview was conducted to assess acceptability and feasibility of the EMA paradigm regarding duration, frequency and items.

The study was approved by the local ethics committee and is in accordance with the Declaration of Helsinki.

## Statistical analysis

Statistical analyses were conducted in R, version 3.4.3 [58]. To test if the content-related and process-related model fit the observed data, we conducted multilevel confirmatory factor analyses (MCFA) starting with the pre-determined sets of items. Model fit was estimated using the R-package lavaan [59], following the procedure described in Huang [60]. As our data had a two-level nested structure (i.e., the occasion level nested into the person level), we assumed the same unifactor structure at each of the levels. A MCFA model is typically specified on decomposed within- and between-person covariance matrices, which allows us to consider both levels simultaneously in a single model.

Two approaches were investigated, one using all four content-related items (i.e., thinking about feelings, problems, past-oriented rumination, future-oriented worry). The second approach comprised the process-related items assessing repetitiveness, intrusiveness, uncontrollability, and burden of RNT. One item assessing duration of RNT was excluded from MCFA analyses based on participant feedback, which indicated that participants had difficulties with answering this item. For each model, goodness-of-fit indices were computed to test whether the model (representing a latent concept of RNT each at the between- and within-person level) fits the data. Goodness-of-fit was evaluated based on the following cutoff criteria: Comparative-Fit Index (CFI) $\geq$ .95 Root Mean Square Error of Approximation (RMSEA) $\leq$ .06, Standardized Mean Square Error of Approximation (SRMR) $\leq$ .08 [61].

Reliability coefficients for multilevel data were calculated for the yielded EMA scale(s) following the generalizability theory (GT) approach [62,63]. Whereas classical test theory decomposes the variance of a given observed score into a true variance and an error variance, GT takes into account several different sources of variance, allowing for the estimation of the reliability of within-person changes (called $R_C$). Furthermore, the GT framework can be used to estimate the reliability of a scale on a randomly selected day ($R_{IR}$) and the reliability of a scale across all days ($R_{KF}$), i.e., the between-person reliability.

To optimize the EMA paradigm in terms of the sampling design, different subsets of data were created for different frequencies (number of notifications per day) and durations (number of days), starting with the lowest acceptable frequency (3 notifications per day) and the lowest acceptable duration (3 days) and ending with the highest possible frequency and duration (i.e., 8 notifications per day for 14 days), which constituted the total set of observations. For frequencies lower than 8, we tried to choose notifications that were equally distributed throughout the day, i.e., for a frequency of 3 notifications per day, we chose to include the 1st, 4th, and 7th notification of the day into the subset. For durations lower than 14 days, we chose a range of subsequent days, always starting from day 1.

Three times per day was determined as the lowest acceptable frequency to account for variations in RNT within the day. Also, a duration of three days was determined as the minimum number to account for day-to-day variations and to increase reliability of daily behavior.

Correlations were calculated between each subset of observations and the total set of observations. Correlations were calculated for different person-level parameters, such as mean, variability, and instability (root mean square successive difference; RMSSD). This resulted in a 12 (number of days: 3–14) x 6 (frequency per day: 3–8) matrix for each parameter. Data were visually inspected to find a reasonable balance between information gain and participant burden, while also accounting for participant feedback from a semi-structured interview at the end of the study to consider acceptability.

## Results

### Data cleaning and compliance

Out of 15,680 possible observations (140 persons x 14 days x 8 assessments per day), 11,673 (76.6%) were completed by participants. Ninety-seven observations (i.e., < 0.01% of total) were deleted due to technical errors. All participants with a response rate of less than 40% were excluded (n = 9). For plausibility, person-level standard deviations were checked for each item across the EMA period. Participants with a standard deviation of 0 in at least one RNT item were excluded (n = 11). Therefore, 120 participants aged 18 to 40 years ($M$ = 22.25, $SD$ = 3.89, 71% female, n = 118) with 10,498 observations remained for data analysis.

### Descriptive data of EMA items

EMA scores were first aggregated at the person level, which were further used to calculate descriptive statistics (see Table 2). Grand means for RNT items ranged from 1.81 for thinking about upsetting memories (PAST), to 2.25 for thinking about negative future situations (PROB). This is comparable to other non-clinical samples [14].

Inspection of the correlation between different RNT items showed that there were high intercorrelations between the content-related items (FEEL, PROB, PAST, FUTR), with the highest correlation between thinking about problems (PROB) and thinking about negative future situations (FUTR; $r$ = .92) and the lowest correlation between thinking about negative future situations (FUTR) and thinking about upsetting memories (PAST; $r$ = .71) on a between-person level (see Table 3). Very high correlations on a between-person level were found between the three PTQ items that capture RNT processes (RPT, INTR, CTRL), ranging from $r$ = .93 to .95. These items also correlated highly on a within-person level, ranging from $r$ = .69 to .70.

**Table 2. Grand means of EMA items.**

| Name | M | SD | Mdn | Min | Max | Range |
|------|------|------|------|------|------|-------|
| 1. FEEL | 2.15 | 0.75 | 2.14 | 1.03 | 4.15 | 3.12 |
| 2. PROB | 2.23 | 0.80 | 2.09 | 1.03 | 4.04 | 3.01 |
| 3. PAST | 1.81 | 0.68 | 1.63 | 1.01 | 3.72 | 2.71 |
| 4. FUTR | 2.25 | 0.89 | 2.08 | 1.03 | 4.73 | 3.70 |
| 5. DUR | 2.00 | 0.73 | 1.84 | 1.01 | 4.33 | 3.32 |
| 6. BUR | 1.96 | 0.75 | 1.78 | 1.01 | 4.08 | 3.07 |
| 7. RPT | 2.16 | 0.95 | 1.97 | 1.01 | 5.02 | 4.01 |
| 8. INTR | 2.06 | 0.98 | 1.72 | 1.01 | 4.94 | 3.93 |
| 9. CTRL | 2.20 | 1.06 | 1.93 | 1.01 | 5.83 | 4.82 |

N = 120; FEEL: feelings; PROB: problems; PAST: upsetting memories; FUTR: negative future situations; DUR: duration; BUR: subjective burden; RPT: repetitiveness; INTR: intrusiveness; CTRL: uncontrollability; EMA: Ecological Momentary Assessment.

**Table 3. Between- and within-person correlations of RNT items.**

|  | 1. FEEL | 2. PROB | 3. PAST | 4. FUTR | 5. DUR | 6. BUR | 7. RPT | 8. INTR | 9. CTRL |
|---|---|---|---|---|---|---|---|---|---|
| 1. FEEL | 1.00 | 0.88 | 0.80 | 0.78 | 0.75 | 0.89 | 0.70 | 0.67 | 0.63 |
| 2 PROB | 0.56 | 1.00 | 0.76 | 0.92 | 0.77 | 0.90 | 0.74 | 0.68 | 0.68 |
| 3. PAST | 0.54 | 0.50 | 1.00 | 0.71 | 0.65 | 0.81 | 0.63 | 0.64 | 0.57 |
| 4. FUTR | 0.49 | 0.68 | 0.47 | 1.00 | 0.72 | 0.86 | 0.74 | 0.67 | 0.68 |
| 5. DUR | 0.46 | 0.52 | 0.44 | 0.51 | 1.00 | 0.75 | 0.58 | 0.51 | 0.50 |
| 6. BUR | 0.57 | 0.65 | 0.51 | 0.62 | 0.53 | 1.00 | 0.78 | 0.73 | 0.74 |
| 7. RPT | 0.53 | 0.55 | 0.47 | 0.54 | 0.50 | 0.62 | 1.00 | 0.95 | 0.95 |
| 8. INTR | 0.49 | 0.52 | 0.46 | 0.52 | 0.46 | 0.62 | 0.69 | 1.00 | 0.93 |
| 9. CTRL | 0.50 | 0.57 | 0.47 | 0.55 | 0.48 | 0.62 | 0.71 | 0.70 | 1.00 |

Above the diagonal = between-person correlations; below the diagonal = within-person correlations; n(within) = 10355, n(between) = 120; FEEL: feelings; PROB: problems; PAST: upsetting memories; FUTR: negative future situations; DUR: duration; BUR: subjective burden; RPT: repetitiveness; INTR: intrusiveness; CTRL: uncontrollability; RNT: repetitive negative thinking.

## Testing different RNT models

Based on the two approaches of selecting the items for our scale, two different models were tested comprising four items each: (a) a content-related approach, including the commonly used items of ruminative self-focus (FEEL, PROB), extended by two items that reflect the temporal dimension (PAST, FUTR) and (b) a process-related approach including the three process variables adapted from the PTQ (RPT, INTR, CTRL), extended by subjective burden/distress regarding RNT (BUR).

**a) Content-related model and hybrid model.** The model including all four content-related items yielded an unacceptable model fit (RMSEA = .174, see Table 4). To explore whether a satisfactory model could be found that retained the two original content items (FEEL, PROB), six different 4-item hybrid models were tested retaining FEEL and PROB in each of the models and testing all combinations with the remaining four process-related items. Two models were found to exhibit acceptable model fit: one model encompassing subjective burden (BUR) and uncontrollability of RNT (CTRL) (RMSEA = .047), another model encompassing subjective burden (BUR) and repetitiveness of RNT (RPT) (RMSEA = .059) in addition to both content items. We chose the former model including CTRL for further analyses, as it yielded the best model fit. MCFA results for all tested hybrid models are shown in S1 Table.

**b) Process-related model.** The second approach included all three adapted PTQ items which assessed core processes of RNT plus a single item to assess subjective burden of RNT. The model showed excellent model fit (RMSEA = .017).

Further analyses were conducted regarding the two scales with good model fit, i.e. the process-related and the exploratory hybrid model.

## Psychometric information

**Reliability.** Between-person reliability for both the process-related and hybrid model was excellent with $R_{KF} > .99$. When looking at within-person reliability, the hybrid model also showed a good reliability of $R_C = .84$; which was lower than the reliability of the process model with $R_C = .88$. When calculating reliability for a random day, reliability was $R_{1R} = .44$ for the hybrid model and $R_{1R} = .50$ for the process-related model.

**Convergent validity.** Small to moderate correlations were found between RNT assessed via EMA and trait questionnaires assessing rumination, worry, and RNT (see Table 5).

**Table 4. Multilevel confirmatory factor analysis results for a content-related, a process-related, and an exploratory hybrid model.**

| | $\chi^2$ | *df* | CFI | SRMR | RMSEA | 90% CI$_L$[1] | 90% CI$_U$[1] |
|---|---|---|---|---|---|---|---|
| **a) Content-related model** | | | | | | | |
| Original model | | | | | | | |
| FEEL, PROB, PAST, FUTR | 636.01*** | 4 | .960 | .040 | .174 | .162 | .185 |
| | W: 610.30 | | | | | | |
| | B: 25.71 | | | | | | |
| Exploratory hybrid model | | | | | | | |
| **FEEL, PROB, BUR, CTRL** | **50.10***** | **4** | **.997** | **.009** | **.047** | **.036** | **.059** |
| | **W: 40.57** | | | | | | |
| | **B: 9.53** | | | | | | |
| **b) Process-related model** | | | | | | | |
| Original model | | | | | | | |
| **RPT, INTR, CTRL, BUR** | **9.82***** | **4** | **1** | **.003** | **.017** | **.003** | **.030** |
| | **W: 7.41** | | | | | | |
| | **B: 2.40** | | | | | | |

[1]Upper/lower confidence interval for RMSEA;

*** $p < .001$;

* $p < .05$;

n(within) = 10355, n(between) = 120. FEEL: feelings; PROB: problems; PAST: upsetting memories; FUTR: negative future situations; BUR: subjective burden; RPT: repetitiveness; INTR: intrusiveness; CTRL: uncontrollability; CFI: Comparative-Fit Index; SRMR: Standardized Mean Square Error of Approximation; RMSEA: Root Mean Square Error of Approximation; W: within-person level; B: between-person level. Final models are highlighted in bold.

Similarly, small to moderate correlations were found between the two EMA scales and symptom scores at baseline.

Furthermore, regression analyses were conducted to inspect associations between EMA-assessed RNT and symptom scores at follow-up. Symptoms of depression, anxiety, and stress after two weeks were significantly predicted by both EMA scales, after controlling for baseline scores for depression, anxiety, and stress, respectively (see Table 6). This finding indicates that both scales have high concurrent validity.

## Sampling design of the EMA paradigm

Correlations of person means between subsets of observations varying in sampling design (frequency of daily assessments, duration of EMA phase) and all observations were calculated for both EMA scales using different statistical values (i.e., mean, standard deviation, and RMSSD).

**Table 5. Correlations between EMA scales, trait measures, and symptom measures.**

| | Trait measures | | | Psychopathology | | | | | |
|---|---|---|---|---|---|---|---|---|---|
| | **PSWQ** | **RSQ-b** | **PTQ** | **PHQ-D** | **GAD-7** | **PSS** | **DASS-D** | **DASS-A** | **DASS-S** |
| EMA-RNT (hybrid) | .32*** | .20* | .36*** | .31*** | .38*** | .36*** | .38*** | .34*** | .24* |
| EMA-RNT (process) | .30*** | .16 | .37*** | .30*** | .33*** | .29*** | .32*** | .37*** | .24* |

*** $p < .001$;

* $p < .05$;

N = 118; EMA; ecological momentary assessment; RNT: repetitive negative thinking; PSWQ: Penn-State Worry Questionnaire; RSQ-b: Response Styles Questionnaire—brooding; PTQ: Perseverative Thinking Questionnaire; PHQ-D: Patient Health Questionnaire—Depression; GAD-7: Generalized Anxiety Disorder Questionnaire; PSS: Perceived Stress Scale; DASS: Depression-Anxiety-Stress-Scales.

**Table 6. Prediction of depression, anxiety, and stress symptoms by the two EMA scales.**

| | Depression Post (DASS) | | | | Anxiety Post (DASS) | | | | Stress Post (DASS) | | | |
|---|---|---|---|---|---|---|---|---|---|---|---|---|
| | *B* | *SE* | β | 95% CI | *B* | *SE* | β | 95% CI | *B* | *SE* | β | 95% CI |
| a) Hybrid model | | | | | | | | | | | | |
| Intercept | -0.44 | .64 | 0.00 | -1.71–0.83 | -0.89 | .43 | 0.00* | -1.74 --0.05 | -0.57 | .79 | 0.00 | -2.13–0.99 |
| Baseline symptoms | 0.48 | .08 | 0.49*** | 0.32–0.64 | 0.57 | .06 | 0.62*** | 0.45–0.69 | 0.53 | .08 | 0.52*** | 0.38–0.68 |
| EMA-RNT (hybrid) | 0.22 | .08 | 0.23** | 0.06–0.37 | 0.20 | .05 | 0.27*** | 0.10–0.31 | 0.29 | .09 | 0.26** | 0.12–0.47 |
| | *Adjusted $R^2$ = .37, F(2,109) = 33.65**** | | | | *Adjusted $R^2$ = .57, F(2,109) = 73.48**** | | | | *Adjusted $R^2$ = .40, F(2,109) = 37.69**** | | | |
| b) Process-related model | | | | | | | | | | | | |
| Intercept | 0.08 | .57 | 0.00 | -1.04–1.20 | -0.56 | .37 | 0.00 | -1.30–0.18 | 0.32 | .72 | 0.00 | -1.10–1.75 |
| Baseline symptoms | 0.51 | .08 | 0.52*** | 0.36–0.67 | 0.57 | .06 | 0.62*** | 0.44–0.69 | 0.55 | .08 | 0.54*** | 0.39–0.71 |
| EMA-RNT (process) | 0.15 | .07 | 0.19* | 0.01–0.28 | 0.17 | .05 | 0.25*** | 0.08–0.26 | 0.18 | .08 | 0.18* | 0.02–0.33 |
| | *Adjusted $R^2$ = .35, F(2,109) = 31.52**** | | | | *Adjusted $R^2$ = .56, F(2,109) = 71.50**** | | | | *Adjusted $R^2$ = .37, F(2,109) = 33.11**** | | | |

*** $p < .001$;

** $p < .01$;

* $p < .05$;

DASS: Depression-Anxiety-Stress-Scales; EMA: ecological momentary assessment; RNT: repetitive negative thinking.

Correlations of person means are shown in Fig 1 for the hybrid EMA scale and in Fig 2 for the process-related EMA scale. Correlations of all person-level parameters are reported in S1 File.

At the maximum frequency of 8 daily assessments, the correlation between person means estimated on the subset and total set of observations exceeded .95 after 8 days. This extent of correlation between person means was reached for frequencies higher than 3 times per day at an EMA duration of at least 10 days across both scales. That is, a subset of observations with a sampling design of 4 notifications per day and a duration of 10 days explains sufficient variance of person means based on all observations.

Considering that about 20% of notifications are missed by participants, a duration of 10 days with a frequency of 5 times per day was chosen as an optimal tradeoff between participant burden and information gain. Correlations between subset and total set of observations ranged from $r = .82$ to .90 for person-level variability and instability across both scales when sampling frequency was fixed at 4 times per day for 10 days (i.e. a sampling design of 5 times per day for 10 days assuming 20% of missing information).

This result was further endorsed by participant feedback, indicating that the frequency during the day appears to exert higher burden than the total duration of days: while 53.5% stated that the frequency of assessments was too high, only 13.2% indicated that the assessment duration of 14 days was too long. According to participant feedback, a frequency of 4 to 5 times per day was perceived as acceptable: 66.6% of participants who answered this question preferred a frequency of 5 or 6 times per day, while another 23.3% of participants stated that 4 times per day was their favored frequency.

## Discussion

The aim of this paper was to develop a reliable EMA paradigm to assess RNT in daily life. To achieve this, we followed three steps: (1) determining the items to be included in the EMA paradigm, (2) examining reliability and construct validity of the EMA scale(s), and (3) optimizing the sampling design while taking into account participant burden.

To determine the optimal combination of items included in the EMA paradigm, we tested a content-related and a process-related scale. Whereas the process-related model showed

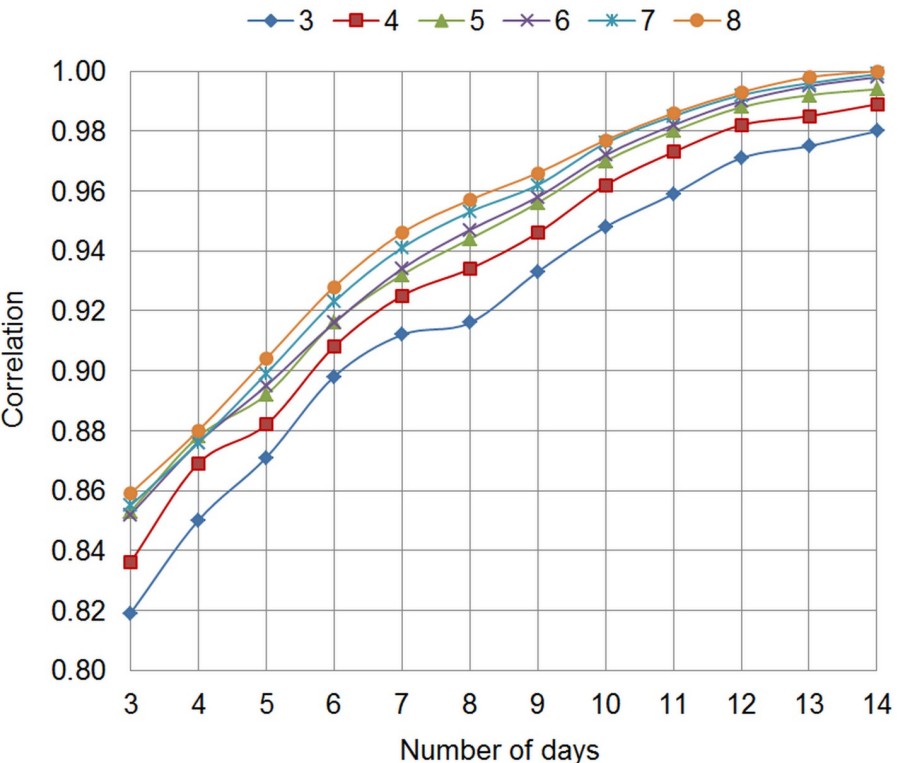

**Fig 1. Hybrid EMA scale.** Correlations of person means between subsets of observations and all observations, varying in frequency between 3 and 8 times per day and duration between 3 and 14 days; EMA: ecological momentary assessment.

excellent model fit according to MCFA, the content-related model yielded unacceptable model fit. To test, whether an acceptable model could be found retaining the two original content items of momentary thinking about problems and feelings, we explored different 4-item hybrid models, each including both original content items and two process-related items. An exploratory hybrid model including difficulty to disengage from thoughts (uncontrollability) and subjective burden (distress) of RNT was found to yield the best model fit.

Next, we investigated reliability and convergent validity of both scales. Both, the process-related and hybrid scale showed good reliability on a between-person level as well as a within-person level, with the process-related scale exhibiting slightly higher reliability. Both scales showed low to moderate associations with standard trait questionnaires on rumination, worry, RNT, and symptom levels of depression, anxiety, and stress. Low to moderate correlations between EMA-assessed rumination or worry and trait measures have also been reported by other studies [14,20,64]. A reason for low correlations between trait measures and daily life RNT might be the different methods in assessing RNT. Whereas EMA aims to reduce retrospective biases and increase ecological validity by assessing RNT in the natural setting in near real-time, trait measures might capture more generalized (metacognitive) beliefs about RNT, rather than the behavior itself.

Additionally, we found that both scales were able to significantly predict symptoms of depression, anxiety, and stress at follow-up over and above baseline symptoms. Since the DASS refers to symptoms within the past week and the follow-up assessment took place shortly after the two week EMA phase, there is an overlap in the time window measured by

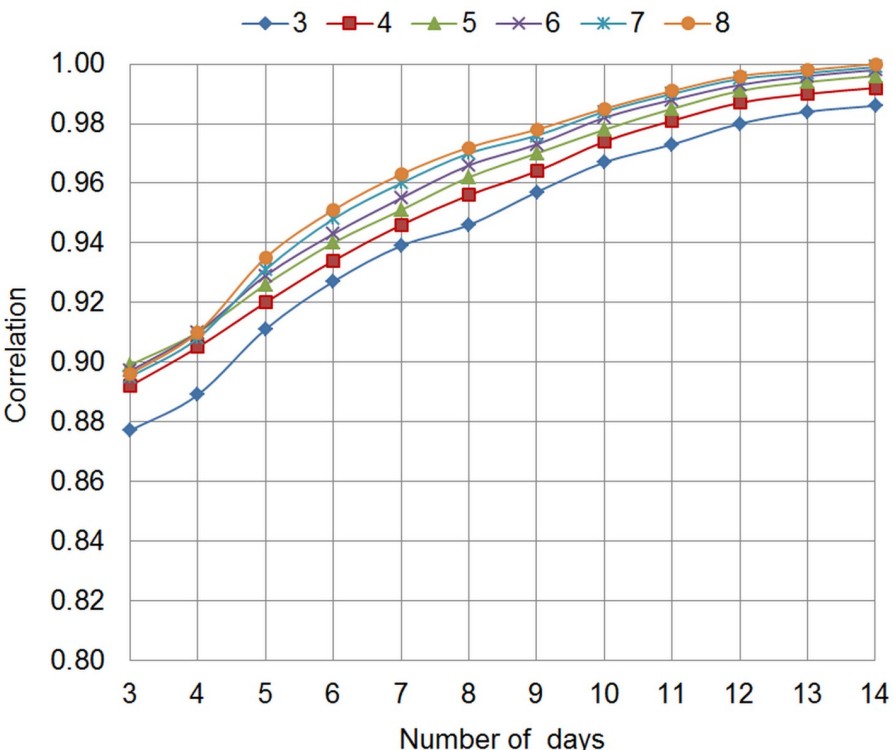

**Fig 2. Process-related EMA scale.** Correlations of person means between subsets of observations and all observations, varying in frequency between 3 and 8 times per day and duration between 3 and 14 days; EMA: ecological momentary assessment.

EMA and by the retrospective DASS. Future studies are needed to determine whether measuring RNT via EMA can significantly predict these symptoms after a longer time interval and whether it outperforms traditional trait questionnaires measuring RNT.

To determine the optimal number of assessment days and frequency of assessments per day, we both analyzed participant feedback based on a semi-structured interview at the end of the EMA phase and computed correlations between subsets of observations and all observations for different sampling designs. A good tradeoff between participant burden and information gain was found at a frequency of five daily assessments for ten consecutive days, allowing for a rate of 20% missing data. Generally, we found that about 40 to 50 observations per participant seemed sufficient to capture the variability and average score of RNT. That is, this number of observations appeared to adequately approximate the person means estimated from all observations.

In sum, the findings of the current paper suggest two possible 4-item EMA scales to assess RNT in daily life, which showed excellent model fit, high reliability within and across persons, as well as significant associations with symptom levels of depression, anxiety, and stress. Both scales share two process-related items, namely, distress (subjective burden of RNT) and uncontrollability (difficulty to disengage from thoughts). It may be that distress/burden and uncontrollability are associated with RNT being experienced negatively. For instance, another study showed that unpleasantness and uncontrollability of RNT were particularly strongly associated with negative affect [52]. Uncontrollability was further associated with longer latencies to fall asleep [53]. Although subjective burden of RNT has not received much interest in previous studies, it seems to be of high importance, as it reflects psychological strain and

distress, which is a clinically relevant aspect across mental disorders. The only difference between both scales was a purely process-related focus versus an integrated approach using both process- and content-related items. Future studies need to cross-validate findings to investigate whether both scales remain to show high reliability and good model fit in other samples or whether one scale exhibits superiority over the other.

To the best of our knowledge, this is the first study to establish an EMA paradigm measuring RNT from a transdiagnostic perspective. Previous studies have either focused on rumination in the context of depression or on worry in the context of anxiety disorders. However, recent research has shown that both rumination and worry share the same underlying process [51]. In order to assess RNT transdiagnostically, it therefore appears important to include items assessing the process of RNT. Of note, studies examining the differences between content-related versus process-related trait questionnaires regarding their predictive utility for depression and anxiety showed that an underlying RNT factor explained more variance in predicting anxiety and depression than disorder-specific cognitive content (i.e., rumination or worry) [65]. It remains to be tested whether the same applies to RNT assessed via EMA. Future studies should compare the two scales identified in the current study regarding the power to predict future symptom levels. As the two scales differ regarding their inclusion of content-related items, this will not only be informative from a purely psychometric but also from a theoretical point of view.

Furthermore, we believe that this is the first study closely investigating the psychometric properties of EMA items to assess RNT. While some previous studies have in part reported reliabilities or concurrent validity of their used scale, unfortunately, this seems to be rather the exception than the rule. Thus, leading researchers in the EMA field are now calling for common standards in study design and selection of items to improve quality of EMA studies and facilitate replication [22].

On the other hand, some limitations are noteworthy. First, this study was not able to determine the predictive validity of the developed EMA paradigm. Consistent with findings of studies assessing RNT via trait questionnaires, a next step in validating the present EMA paradigm is to investigate whether EMA-assessed RNT significantly predicts symptoms of anxiety and depression after a longer time interval. In addition to examining whether an aggregated score of RNT across the EMA phase significantly predicts psychopathology, other dynamic measures can be taken into account. For instance, one study has shown that EMA-assessed instability of rumination predicts depressive symptoms after six and 36 months over and above average levels of daily-life rumination [66]. Whereas the underlying causes for this finding remain to be investigated, the authors speculated that daily life stressors may be related to higher fluctuations in rumination, resulting in higher depression levels. Therefore, not only instability of RNT, but also stressors and the interaction of RNT with stress in daily life might be valuable targets for further investigation [67].

Furthermore, our proposed frequency and duration of the EMA paradigm has to be cross-validated in a future study, since our findings are based on post-hoc analyses without actually comparing different frequencies and durations in vivo. Moreover, the sampling design of EMA studies is always dependent on the base rate or temporal variability of the measured construct. Therefore, our finding cannot be generalized to EMA studies overall, but is confined to the measurement of RNT.

Last, we assessed RNT in a non-clinical sample. Therefore, we excluded participants which indicated to be suffering from a mental disorder. Inspection of the GAD and depression scores showed that 15 participants scored in the range for clinically significant levels of generalized anxiety ($\geq 10$), while 13 participants exceeded the cutoff score ($\geq 10$) for moderate levels of depressive symptoms. In general, the majority of the sample can be labeled as healthy

participants, which is also reflected in the low mean scores of RNT. The results of this paper are, therefore, confined to non-clinical populations and further research is necessary to investigate whether the same paradigm can be replicated in clinical populations.

Nevertheless, this study proposes two promising EMA scales for assessing RNT in daily life, which are subject to further investigations. With this study, we hope to encourage other researchers to also consider a process-oriented, transdiagnostic perspective on rumination and worry.

## Supporting information

**S1 Table. Multilevel confirmatory factor analysis results for different exploratory hybrid models.**
(DOCX)

**S1 File. Correlations between subsets of observations and all observations for varying sampling designs regarding the hybrid EMA scale and the process-related EMA scale.**
(XLSX)

## Acknowledgments

The authors would like to thank Charlotte Ebrecht and other students for their support in data collection.

## Author Contributions

**Conceptualization:** Tabea Rosenkranz, Edward R. Watkins, Thomas Ehring.

**Data curation:** Tabea Rosenkranz, Keisuke Takano.

**Formal analysis:** Tabea Rosenkranz, Keisuke Takano.

**Funding acquisition:** Edward R. Watkins, Thomas Ehring.

**Investigation:** Tabea Rosenkranz.

**Methodology:** Tabea Rosenkranz, Keisuke Takano.

**Project administration:** Tabea Rosenkranz, Thomas Ehring.

**Resources:** Thomas Ehring.

**Supervision:** Tabea Rosenkranz, Thomas Ehring.

**Visualization:** Tabea Rosenkranz.

**Writing – original draft:** Tabea Rosenkranz.

**Writing – review & editing:** Tabea Rosenkranz, Keisuke Takano, Edward R. Watkins, Thomas Ehring.

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
