## [Decision Letter · Decision Letter 0]

22 Jan 2020

PONE-D-19-32809

Assessing repetitive negative thinking in daily life: Development of an ecological momentary assessment paradigm

PLOS ONE

Dear Dr Rosenkranz,

Thank you for submitting your manuscript to PLOS ONE. I was fortunate to have an expert in the field review this manuscript. I have had some difficulty finding a second reviewer, however the first reviewer's review was thorough enough that I am able to make a decision based on that. As you will see below, the reviewer found your paper to have much merit, especially in the novel EMA design. Despite this, the reviewer also found some issues with the paper that are addressable in a revision. Therefore, we invite you to submit a revised version of the manuscript that addresses the points raised during the review process.

The full reviews are pasted below my signature and as you will see, many of the comments are in regard to the need to provide more clarity around your participant selection process and reduce some redundancy in the introduction. 

We would appreciate receiving your revised manuscript by Mar 07 2020 11:59PM. To enhance the reproducibility of your results, we recommend that if applicable you deposit your laboratory protocols in protocols.io, where a protocol can be assigned its own identifier (DOI) such that it can be cited independently in the future. For instructions see: http://journals.plos.org/plosone/s/submission-guidelines#loc-laboratory-protocols

We look forward to receiving your revised manuscript.

Kind regards,

Evan M Kleiman

Academic Editor

PLOS ONE

2. We note that Table 2 and File S1 in the Supporting Information may include questionnaire items that may have been previously published. The reproduction of previously published work has implications for the copyright that may apply to these publications. We would be grateful if you could clarify whether you have obtained permission from the original copyright holder to republish these items under a CC BY license. If you have not obtained permission to publish these items please remove them from your manuscript. You may wish to replace the text you have removed with relevant question numbers/ brief descriptions of each item; please be sure to include any relevant references and in-text citations.

Reviewers' comments:

Reviewer's Responses to Questions

**Comments to the Author**

1. Is the manuscript technically sound, and do the data support the conclusions?

Reviewer #1: Yes

2. Has the statistical analysis been performed appropriately and rigorously? 

Reviewer #1: I Don't Know

3. Have the authors made all data underlying the findings in their manuscript fully available?

Reviewer #1: Yes

4. Is the manuscript presented in an intelligible fashion and written in standard English?

Reviewer #1: Yes

5. Review Comments to the Author

Reviewer #1: This is a well-written and thorough paper attempting to establish a psychometrically-sound, transdiagnostic EMA measure of repetitive negative thinking. It is an important addition to the literature, as the authors rightfully point out that most EMA studies are using untested questionnaires- therefore, moving forward, researchers could employ the items described by the authors and be able to cite the work they have done here to establish reliability/validity/model fit.

My concerns with the manuscript in its current form are as follows:

1. Introduction: Much of the info at the end of the introduction gets repeated in the methods section. It seems to belong fully in the methods and can be cut from the intro. The introduction should frame the question at hand, which it does very nicely up to line 90 on p. 4. From line 91 on, you go into a fairly detailed account of your methods. I would work on cutting a lot of this out/moving info to the methods section if it is not already described there. Retaining big picture concepts (e.g. your discussion of content vs process) is important, but description of item-level info can be saved for the methods.

2. Participants: participants were excluded if they were in treatment for a mental disorder. Please note why this was an exclusionary criterion. Were you attempting to have a non-clinical sample? As we know rates of mental health treatment are low, it is possible that some in your sample still will have scored within clinical ranges on measures of anxiety/depression (eg GAD 7, PHQ D)- please report this information. The attempt to exclude more symptomatic participants in the current study should also be noted in the limitations/future directions section of your discussion- the means on your EMA items seem fairly low across the board, so it is worth noting whether your findings may have been different, including model fit, in a more clinical sample.

3. “Ecological Momentary Assessment”, p. 9 says the 12 hour window on weekends started at 12AM- should this read PM?

4. Statistical analysis: I have not used MCFA so cannot comment on the appropriateness/conduct of these analyses, but feel that this section is well-written

5. P.13. I’m still a little fuzzy on how you calculated subsets. Did you randomly select a given # of responses per day for a participant? And was it only for some days out of the full 14 day period? I would suggest rewording and maybe asking someone unfamiliar with the study to read this section and see if they understand the procedure.

6. Tables 3 and 4- nb and nw phrasing may be confusing to the reader, I suggest spelling out nbetween, nwithin

7. Table 6- looks like the DASS section doesn’t have the beta symbol and instead has a B

8. Discussion: “low to moderate correlations between EMA-assessed measures & trait measures have been reported by other studies.” Expand a bit here on why this might be the case- do you think state measures are more valid? You bring this up in the intro, but don’t bring it back into the discussion.

9. This study finds that the process items tended to fare better than the content items. What might this mean? For instance, you write “Therefore, adding process-related variables to ensure a content independent measurement of RNT in daily life is crucial to investigate RNT as a transdiagnostic process across mental disorders.” Does this mean that ideally content-related items shouldn’t be included at all? More discussion of the implications of your findings would be helpful in the discussion.

10. “In addition to examining whether an aggregated score of RNT across the EMA phase significantly predicts psychopathology, other dynamic measures can be taken into account, such as instability or stress-reactive RNT.” Flesh this idea out a bit more- I didn’t understand how the second half of the sentence related to the first.

11. Supporting table 2 shows that FEEL, PROB, RPT, BUR has an RMSEA of .059. I am not expert in RMSEA but read that values below .06 indicated good fit. In which case, should this be noted in the body of your paper?

6. PLOS authors have the option to publish the peer review history of their article (what does this mean?). If published, this will include your full peer review and any attached files.

Reviewer #1: No

---

## [Author Response · Author response to Decision Letter 0]

7 Mar 2020

We would like to thank the editor and the reviewer for their thoughtful comments, which have helped us improve the manuscript. We have carefully checked every comment and hope to have answered your comments in a satisfactory manner. Please find our rebuttal below.

RESPONSE: We have carefully reviewed the style requirements and have adapted our style where needed.

2. We note that Table 2 and File S1 in the Supporting Information may include questionnaire items that may have been previously published. The reproduction of previously published work has implications for the copyright that may apply to these publications. We would be grateful if you could clarify whether you have obtained permission from the original copyright holder to republish these items under a CC BY license. If you have not obtained permission to publish these items please remove them from your manuscript. You may wish to replace the text you have removed with relevant question numbers/ brief descriptions of each item; please be sure to include any relevant references and in-text citations.

RESPONSE: S1 has now been removed. The RNT items are provided in Table 1 within the manuscript. All other items used during the EMA assessment are briefly described within the text. 

As to the RNT items, we have obtained permission to republish the items for Duration, Burden/Distress, Repetitiveness, Intrusiveness, and Uncontrollability. The items for RNT focused on feelings, problems, past, and future were modelled after items used in earlier studies (as described in the manuscript), but did not use the exact phrasing used in these earlier studies. This means that we are not republishing content from earlier manuscripts in these cases. We have now made this aspect clearer in the manuscript. 

Reviewers' comments:

Reviewer's Responses to Questions

Comments to the Author

1. Is the manuscript technically sound, and do the data support the conclusions?

Reviewer #1: Yes

2. Has the statistical analysis been performed appropriately and rigorously?

Reviewer #1: I Don't Know

3. Have the authors made all data underlying the findings in their manuscript fully available?

Reviewer #1: Yes

4. Is the manuscript presented in an intelligible fashion and written in standard English?

Reviewer #1: Yes

5. Review Comments to the Author

Reviewer #1: This is a well-written and thorough paper attempting to establish a psychometrically-sound, transdiagnostic EMA measure of repetitive negative thinking. It is an important addition to the literature, as the authors rightfully point out that most EMA studies are using untested questionnaires- therefore, moving forward, researchers could employ the items described by the authors and be able to cite the work they have done here to establish reliability/validity/model fit.

My concerns with the manuscript in its current form are as follows:

1. Introduction: Much of the info at the end of the introduction gets repeated in the methods section. It seems to belong fully in the methods and can be cut from the intro. The introduction should frame the question at hand, which it does very nicely up to line 90 on p. 4. From line 91 on, you go into a fairly detailed account of your methods. I would work on cutting a lot of this out/moving info to the methods section if it is not already described there. Retaining big picture concepts (e.g. your discussion of content vs process) is important, but description of item-level info can be saved for the methods.

RESPONSE: Thank you for this comment. We have now merged parts of the detailed introduction after line 91 with the EMA item description in methods. As a result, we believe that the introduction is now more straightforward and focuses on presenting the bigger picture.

2. Participants: participants were excluded if they were in treatment for a mental disorder. Please note why this was an exclusionary criterion. Were you attempting to have a non-clinical sample? As we know rates of mental health treatment are low, it is possible that some in your sample still will have scored within clinical ranges on measures of anxiety/depression (eg GAD 7, PHQ D)- please report this information. The attempt to exclude more symptomatic participants in the current study should also be noted in the limitations/future directions section of your discussion- the means on your EMA items seem fairly low across the board, so it is worth noting whether your findings may have been different, including model fit, in a more clinical sample.

RESPONSE: We aimed for a non-clinical sample because the development of the EMA paradigm was part of a grant-funded project focusing on prevention. This means that within this project our EMA paradigm will be applied to a non-clinical sample. We have added this part in the paper (p. 5): 

Participants were included in the study if they had German language skills comparable to a native speaker and were currently not in treatment for mental disorders, as this study aimed for a non-clinical sample

We have included this issue as a limitation in the discussion part (p. 22): 

Last, we assessed RNT in a non-clinical sample. Therefore, we excluded participants who indicated to be suffering from a mental disorder. Inspection of the GAD and depression scores showed that 15 participants scored in the range for clinically significant levels of generalized anxiety (≥ 10), while 13 participants exceeded the cutoff score (≥ 10) for moderate levels of depressive symptoms. In general, the majority of the sample can be labeled as healthy participants, which is also reflected in the low mean scores of RNT. The results of this paper are, therefore, confined to non-clinical populations and further research is necessary to investigate whether the same paradigm can be replicated in clinical populations. 

3. “Ecological Momentary Assessment”, p. 9 says the 12 hour window on weekends started at 12AM- should this read PM?

RESPONSE: Yes, we have corrected this error.

4. Statistical analysis: I have not used MCFA so cannot comment on the appropriateness/conduct of these analyses, but feel that this section is well-written

RESPONSE: Thank you. As described, the fine details of MCFA can be consulted in the paper by Huang, 2017, which we have cited.

5. P.13. I’m still a little fuzzy on how you calculated subsets. Did you randomly select a given # of responses per day for a participant? And was it only for some days out of the full 14 day period? I would suggest rewording and maybe asking someone unfamiliar with the study to read this section and see if they understand the procedure.

RESPONSE: We have rephrased this paragraph so that the procedure will become clearer for the reader. As described, the number of notifications per day were “equally spaced throughout the day”, i.e., for a frequency of 3 notifications, we would have used e.g. the 1st, 4th, and 7th notification. We have also included this clarification in the manuscript (p. 14).

6. Tables 3 and 4- nb and nw phrasing may be confusing to the reader, I suggest spelling out nbetween, nwithin

RESPONSE: We have changed this in the tables.

7. Table 6- looks like the DASS section doesn’t have the beta symbol and instead has a B

RESPONSE: Thank you for noticing this error. We have corrected it.

8. Discussion: “low to moderate correlations between EMA-assessed measures & trait measures have been reported by other studies.” Expand a bit here on why this might be the case- do you think state measures are more valid? You bring this up in the intro, but don’t bring it back into the discussion.

RESPONSE: We now bring this issue back in the discussion (p. 20):

A reason for low correlations between trait measures and daily life RNT might be the different methods in assessing RNT. Whereas EMA aims to reduce retrospective biases and increase ecological validity by assessing RNT in the natural setting in near real-time, trait measures might capture more generalized (metacognitive) beliefs about RNT, rather than the behavior itself.

9. This study finds that the process items tended to fare better than the content items. What might this mean? For instance, you write “Therefore, adding process-related variables to ensure a content independent measurement of RNT in daily life is crucial to investigate RNT as a transdiagnostic process across mental disorders.” Does this mean that ideally content-related items shouldn’t be included at all? More discussion of the implications of your findings would be helpful in the discussion.

RESPONSE: In the revised manuscript, we now elaborate on this issue. Specifically, we (1) now argue that process-related items need to be included to assess RNT transdiagnostically, (2) refer to earlier research using trait questionnaires showing that process-related measures are better at predicting psychopathology than content-related ones, but (3) acknowledge that it is ultimately an empirical question whether the same applies to assessing RNT using EMA. The paragraph on p. 21 now reads:

To the best of our knowledge, this is the first study to establish an EMA paradigm measuring RNT from a transdiagnostic perspective. Previous studies have either focused on rumination in the context of depression or on worry in the context of anxiety disorders. However, recent research has shown that both rumination and worry share the same underlying process [50]. In order to assess RNT transdiagnostically, it therefore appears important to include items assessing the process of RNT. Of note, studies examining the differences between content-related versus process-related trait questionnaires regarding their predictive utility for depression and anxiety showed that an underlying RNT factor explained more variance in predicting anxiety and depression than disorder-specific cognitive content (i.e., rumination or worry) [64]. It remains to be tested whether the same applies to RNT assessed via EMA. Future studies should compare the two scales identified in the current study regarding the power to predict future symptom levels. As the two scales differ regarding their inclusion of content-related items, this will not only be informative from a purely psychometric but also from a theoretical point of view. 

10. “In addition to examining whether an aggregated score of RNT across the EMA phase significantly predicts psychopathology, other dynamic measures can be taken into account, such as instability or stress-reactive RNT.” Flesh this idea out a bit more- I didn’t understand how the second half of the sentence related to the first.

RESPONSE: We have restructured this part of the discussion and have now added more information. The main message we wanted to get across is the idea that not only mean levels of daily RNT are important predictors of psychopathology, but that we can also exploit the advantages of EMA, i.e. taking into account the temporal dynamics of EMA variables (p. 22):

In addition to examining whether an aggregated score of RNT across the EMA phase significantly predicts psychopathology, other dynamic measures can be taken into account. For instance, one study has shown that EMA-assessed instability of rumination predicts depressive symptoms after six and 36 months over and above average levels of daily-life rumination [65]. Whereas the underlying causes for this finding remain to be investigated, the authors speculated that daily life stressors may be related to higher fluctuations in rumination, resulting in higher depression levels. Therefore, not only instability of RNT, but also stressors and the interaction of RNT with stress in daily life might be valuable targets for further investigation [66]. 

11. Supporting table 2 shows that FEEL, PROB, RPT, BUR has an RMSEA of .059. I am not expert in RMSEA but read that values below .06 indicated good fit. In which case, should this be noted in the body of your paper?

RESPONSE: Indeed, we found two models with acceptable fit, however, we decided to include the model with the best fit in further analyses. We, therefore, rephrased our results to clarify this (p. 14):

Two models were found to exhibit acceptable model fit: one model encompassing subjective burden (BUR) and uncontrollability of RNT (CTRL) (RMSEA = .047), another model encompassing subjective burden (BUR) and repetitiveness of RNT (RPT) (RMSEA = .059) in addition to both content items. We chose the former model including CTRL for further analyses, as it yielded the best model fit.

---

## [Editor Report · Decision Letter 1]

1 Apr 2020

Assessing repetitive negative thinking in daily life: Development of an ecological momentary assessment paradigm

PONE-D-19-32809R1

Dear Dr. Rosenkranz,

We are pleased to inform you that your manuscript has been judged scientifically suitable for publication and will be formally accepted for publication once it complies with all outstanding technical requirements. This paper will make an excellent addition to the literature!

With kind regards,

Evan M Kleiman

Academic Editor

PLOS ONE
---

## [Editor Report · Acceptance letter]

7 Apr 2020

PONE-D-19-32809R1 

Assessing repetitive negative thinking in daily life: Development of an ecological momentary assessment paradigm 

Dear Dr. Rosenkranz:

I am pleased to inform you that your manuscript has been deemed suitable for publication in PLOS ONE. Congratulations! Your manuscript is now with our production department. 

With kind regards,

on behalf of

Dr. Evan M Kleiman 

Academic Editor

PLOS ONE